# The Mamba in the Llama:
# Distilling and Accelerating Hybrid Models

**Junxiong Wang**[1*]   **Daniele Paliotta**[2,3*]   **Avner May**[3]   **Alexander M. Rush**[1]   **Tri Dao**[3,4]

[1]Cornell University   [2]University of Geneva   [3]Together AI   [4]Princeton University

## Abstract

Linear RNN architectures, like Mamba, can be competitive with Transformer models in language modeling while having advantageous deployment characteristics. Given the focus on training large-scale Transformer models, we consider the challenge of converting these pretrained models for deployment. We demonstrate that it is feasible to distill large Transformers into linear RNNs by reusing the linear projection weights from attention layers with academic GPU resources. The resulting hybrid model, which incorporates a quarter of the attention layers, achieves performance comparable to the original Transformer in chat benchmarks and outperforms open-source hybrid Mamba models trained from scratch with trillions of tokens in both chat benchmarks and general benchmarks. Moreover, we introduce a hardware-aware speculative decoding algorithm that accelerates the inference speed of Mamba and hybrid models. Overall we show how, with limited computation resources, we can remove many of the original attention layers and generate from the resulting model more efficiently. Our top-performing model, distilled from Llama3-8B-Instruct, achieves a 29.61 length-controlled win rate on AlpacaEval 2 against GPT-4 and 7.35 on MT-Bench, surpassing the best 8B scale instruction-tuned linear RNN model. We also find that the distilled model has natural length extrapolation, showing almost perfect accuracy in the needle-in-a-haystack test at 20x the distillation length. Code and pre-trained checkpoints are open-sourced at https://github.com/jxiw/MambaInLlama and https://github.com/itsdaniele/speculative_mamba.

## 1   Introduction

While Transformers [73] have been an essential architecture in deep learning and have driven the success of large language models such as GPT [9], Llama [71], and Mistral [37], they are prohibitively slow for very long sequence generation due to their quadratic complexity with respect to sequence length and large key-value (KV) cache requirement. Recent linear RNN models (Mamba [26], Mamba2 [18], GLA [79], RWKV [55], RetNet [68], Griffin [19]) beat Transformers in controlled experiments at small to medium scale, although the best Transformers still significantly outperform these models on downstream tasks. We note that the training times of linear RNN models are similar to those of highly optimized Transformers [79], and therefore scaling up either of these models requires substantial computational resources.

---

[*]Equal Contribution. Order determined by coin flip. Correspondence to: *junxiong@cs.cornell.edu* and *daniele.paliotta@unige.ch*

38th Conference on Neural Information Processing Systems (NeurIPS 2024).

The primary benefit of linear RNN models (Mamba [26], Mamba2 [18]) is that they have faster inference (5× higher throughput) than Transformers. Efficient inference is emerging as a critical need for LLM systems such as new applications currently bottlenecked by the large KV cache of Transformers, e.g. reasoning over multiple long documents [30, 65, 56] and files in large codebases [61, 42]). Emerging workflows with agents [81, 77] also require large-batch inference to explore more trajectories and long-context to model complex environments.

These properties motivate the goal of distilling a large pretrained Transformer model into a linear RNN in order to generate as efficiently as possible. The technical challenges are two-fold: how to map pretrained Transformer weights to linear RNN weights for distillation, and how to adapt best-practice Transformer inference techniques, such as speculative decoding, to the new architecture. We make the following contributions:

- We show that by reusing weights from attention layers, it is possible to distill a large transformer into a large hybrid-linear RNN with minimal additional compute while preserving much of its generation quality. We propose a modified Mamba architecture that can be directly initialized from the attention block of a pretrained model.
- We propose a multistage distillation approach that mirrors the standard LLM pipeline combining progressive distillation, supervised fine-tuning [39], and directed preference optimization [58]. This approach shows better perplexity and downstream evaluation compared with vanilla distillation.
- We develop a hardware-aware speculative sampling algorithm and a fast kernel for speculative decoding on Mamba and hybrid architectures. We achieve a throughput of over 300 tokens/second for a Mamba 7B model. Additionally, we show that speculative decoding can be effectively applied to our hybrid architecture.

Our experiments distill different large-scale open chat LLMs, Zephyr-7B [72], Llama-3 8B [21] to linear RNN models (hybrid Mamba and Mamba2), using only 20B tokens of training. Results show that the distilled approach matches the teacher model in standard Chat benchmarks [84, 43]. We also show that it performs on par or better with all similarly sized pretrained-from-scatch Mamba models including Mamba 7B models [52, 26] trained from scratch with 1.2T tokens or NVIDIA Hybrid Mamba2 models [74] trained from scratch with 3.5T tokens in multiple tasks (e.g., MMLU [34], TruthfulQA [47]) in the LM evaluation [25]. Concurrent with this work, MOHAWK [6] distills a Mamba-2 variant based on the Phi-1.5 architecture with limited computation costs and performance loss.

## 2 From Transformer to Mamba

### 2.1 Relationship Between Attention and Linear RNNs

We begin by reviewing multihead attention to clarify the shapes of intermediate objects. Notationally, we use explicit subscripts for the sequence position instead of matrix representation, to better highlight similarities between the two models.

Attention is computed in parallel for multiple differently parameterized heads. Each head takes sequence $\boldsymbol{o}$ with hidden size $D$ as an argument and computes,

$$\mathbf{Q}_t = \mathbf{W}^Q \boldsymbol{o}_t, \quad \mathbf{K}_t = \mathbf{W}^K \boldsymbol{o}_t, \quad \mathbf{V}_t = \mathbf{W}^V \boldsymbol{o}_t \quad \text{for all } t,$$

$$\alpha_1 \ldots \alpha_T = \text{softmax}\big([m_{1,t}\mathbf{Q}_t^\top \mathbf{K}_1 \ldots m_{T,t}\mathbf{Q}_t^\top \mathbf{K}_T]/\sqrt{D}\big) \quad \boldsymbol{y}_t = \sum_{s=1}^{t} \alpha_s \mathbf{V}_s$$

$$\text{where } \boldsymbol{o}_t \in \mathbb{R}^{D \times 1}, \quad \mathbf{W} \in \mathbb{R}^{N \times D} \quad \mathbf{Q}_t, \mathbf{K}_t, \mathbf{V}_t \in \mathbb{R}^{N \times 1} \quad m_{s,t} = \mathbf{1}(s \leq t)$$

Recent work has argued that linear RNNs can be serious competitors to attention in large language models. Several different linear RNN formulations have been proposed with similar formulations. For now, we leave the shapes of the parameters $\mathbf{A}_t, \mathbf{B}_t, \mathbf{C}_t$ abstract, and note that linear RNNs all take the following form that maps a 1-dimensional sequence to another through an implicit matrix-valued hidden state $\boldsymbol{h}$.

$$\boldsymbol{h}_t = \mathbf{A}_t \boldsymbol{h}_{t-1} + \mathbf{B}_t x_t, \quad y_t = \mathbf{C}_t^\top \boldsymbol{h}_t \tag{1}$$

Linear RNNs have several computational advantages over attention. During training, all $y_t$ values can be computed more efficiently than attention since there is no softmax non-linearity. During inference, each next $y_t$ can be computed serially without requiring a cache.

Despite the superficially different form, there is a natural relationship between linear RNNs and attention. *Linearizing* the attention formula by removing the softmax yields:

$$y_t = \sum_{s=1}^{t} \alpha_s v_s = \frac{1}{\sqrt{D}} \sum_{s=1}^{t} m_{s,t} \mathbf{Q}_t^\top \mathbf{K}_s v_s = \frac{1}{\sqrt{D}} \mathbf{Q}_t^\top \sum_{s=1}^{t} m_{s,t} \mathbf{K}_s v_s$$

This implies that there exists a linear RNN form of linearized attention, specifically:

$$\boldsymbol{h}_t = m_{t-1,t} \boldsymbol{h}_{t-1} + \mathbf{K}_t v_t \quad y_t = \frac{1}{\sqrt{D}} \mathbf{Q}_t^\top \boldsymbol{h}_t$$

$$\downarrow$$

$$\boldsymbol{h}_t = \mathbf{A}_t \boldsymbol{h}_{t-1} + \mathbf{B}_t x_t, \quad y_t = \mathbf{C}_t^\top \boldsymbol{h}_t$$
$$\mathbf{A}_t = m_{t-1,t}, \ \mathbf{B}_t = \mathbf{W}^K \boldsymbol{o}_t, \ \mathbf{C}_t = \mathbf{W}^Q \boldsymbol{o}_t, \ \boldsymbol{x}_t = \mathbf{W}^V \boldsymbol{o}_t$$

Note though that this version uses a hidden state of size $\boldsymbol{h} \in \mathbb{R}^{N \times 1}$. Effectively tracking only one scalar over time per hidden dimension. Naively applying this transformation leads to poor results. The issue is that linearizing attention produces a degraded representation of the original model, as the softmax nonlinearity is critical to attention.

The key to improving these models is to increase the capacity of the linear hidden state to better capture long-term structure. For example, previous work has shown the use of kernel methods to improve this approximation [63, 36, 83]. These approaches expand the size of the hidden state representation to $\boldsymbol{h}$ to $\mathbb{R}^{N \times N'}$ to better match the modeling capacity of softmax.

## 2.2 Distilling to an Expanded Linear RNN

To design a effective distilled linear RNN, we aim to stay as close as possible to the original Transformer parameterization, while also expanding the capacity of the linear RNN in an efficient manner. We will *not* attempt to have the new model capture the exact original attention function, but instead use the linearized form as a starting point for distillation.

Specifically, we adapt the parameterization from Mamba, [26] to increase the hidden state size, while initializing from the attention representation. Mamba uses a continuous time state-space model (SSM) to parameterize a linear RNN at run time, described by the differential equation,

$$\boldsymbol{h}'(k) = \mathbf{A}\boldsymbol{h}(k) + \mathbf{B}(k)\boldsymbol{x}(k) \quad \boldsymbol{y}(k) = \mathbf{C}(k)\boldsymbol{h}(k)$$

Where $\mathbf{A}$ is a diagonal matrix and other values are continuous signals. To apply this formulation to a discrete-time problem like language modeling, we use a neural network to produce a sequence of sampling intervals $\Delta_t$ and samples of the signals at these time steps. Given these sampling intervals, and $T$ samples of $\mathbf{B}, \mathbf{C}$, Mamba approximates the continuous-time equation using a linear RNN as a discretization. We use an overbar to indicate the discrete-time form, which is reconstructed dynamically.

---

**Algorithm 1** Attention-Initialized Mamba

1: **Shapes:** $B$ - Batch, $L$ - Length, $D$ - embed size,
2:       $N = D/$Heads, $N'$ - expand
3: **Input:** $\boldsymbol{o}_t$: (B, D)
4: **Output:** output: (B, D)
5: **New Params:** MLP, $\mathbf{A}$
6: **for** each head $\mathbf{W}^k, \mathbf{W}^q, \mathbf{W}^v, \mathbf{W}^o : (N, D)$
7:       expanding grouped KVs **do**
8:      **Head Parameter:** $\mathbf{A} : (N, N')$
9:      for all positions $t$:
10:       $\boldsymbol{x}_t : (B, N) \leftarrow \mathbf{W}^V \boldsymbol{o}_t$
11:       $\mathbf{B}_t : (B, N) \leftarrow \mathbf{W}^K \boldsymbol{o}_t$
12:       $\mathbf{C}_t : (B, N) \leftarrow \mathbf{W}^Q \boldsymbol{o}_t$
13:       $\Delta_t : (B, N') \leftarrow \text{MLP}(\boldsymbol{x}_t)$
14:       $\overline{\mathbf{A}}_{1:T}, \overline{\mathbf{B}}_{1:T}, \overline{\mathbf{C}}_{1:T} \quad : \quad (B, N, N') \leftarrow \text{DISC}(\mathbf{A}, \mathbf{B}, \mathbf{C}, \Delta)$
15:       $y \leftarrow \text{LINEARRNN}(\overline{\mathbf{A}}, \overline{\mathbf{B}}, \overline{\mathbf{C}}, \boldsymbol{x})$
16:       output $\leftarrow$ output $+ \mathbf{W}^{O\top} y$
17: **return** output

---

$$\overline{\mathbf{A}}_{1\ldots T}, \overline{\mathbf{B}}_{1\ldots T}, \overline{\mathbf{C}}_{1\ldots T} = \text{Discretize}(\mathbf{A}, \mathbf{B}_{1\ldots T}, \mathbf{C}_{1\ldots T}, \Delta_{1\ldots T})$$

In this simplest case, with $N' = 1$ and an identity discretization, this approach recovers the linear attention to linear RNN conversion discussed in the previous section. The benefit of Mamba is that with $N' > 1$ the continuous-time parameterization allows the model to learn significantly richer functions, without many more parameters or decreased efficiency. Specifically the only additional learned parameters will be the sampling rate $\Delta$ and the dynamic $\mathbf{A}$. These new parameters will control the constructed linear RNN through the discretization function yielding the new matrix valued linear RNN. Specifically, we take in the same $\mathbf{B}_t, \mathbf{C}_t \in \mathbb{R}^{N \times 1}$ and $\Delta_t \in \mathbb{R}^{N'}$, but output $\overline{\mathbf{B}}_t, \overline{\mathbf{C}}_t \in \mathbb{R}^{N' \times N \times 1}$, effectively increasing the hidden size by a factor of $N'$ over the naive linear attention.

A core contribution of Mamba [26, 18] is to demonstrate a hardware-aware factorization of this algorithm. Implementing the algorithm naively would be prohibitively slow as the new expanded parameters are quite large. Their approach fuses discretization, state expansion, and applying the linear RNN into a single kernel, which circumvents fully materializing the discrete parameters. This allows for large $N'$ with relatively small efficiency costs.

### 2.3 Attention-to-Mamba Initialization and Hybrid Stepwise Training

Our full approach is shown in Algorithm 1. This algorithm feeds the standard $\mathbf{Q}, \mathbf{K}, \mathbf{V}$ heads from attention directly into the Mamba discretization, and then applies the resulting linear RNN. As noted above, this can seen as roughly initializing with linearized attention and allowing the model to learn richer interactions through the expanded hidden state.

Figure 1 shows the resulting architecture. Our version directly replaces Transformer attention heads directly with fine-tune linear RNN layers. We keep the Transformer MLP layers as is and do not train them. This approach also requires processing additional components like grouped query attention that shares keys and values across heads. We note that this architecture differs from the architecture used in many Mamba systems, which combines MLP-SSM layers and uses a single head.

This initialization allows us to replace any attention block with a linear RNN block. We experiment with *hybrid* models where we keep every $n$ attention layers. Empirically we found that replacing layers in a stepwise manner was the most effective strategy, i.e. we first keep every 2 layers, distill, and then every 4, and continue distillation.

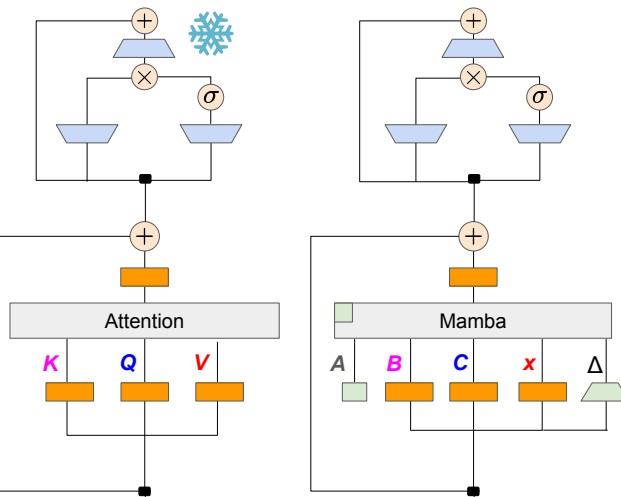

Figure 1: Transferring Transformer to Mamba. Weights, in orange, are initialized from the Transformer (Linear projections for $\mathbf{Q}, \mathbf{K}$, and $\mathbf{V}$ are initialized using linear projection for $\mathbf{C}, \mathbf{B}$, and $\mathbf{X}$ respectively). We replace individual attention heads with Mamba heads, and then finetune Mamba blocks while freezing the MLP blocks. Shapes are kept mainly the same. Weights in green are added. New parameters are introduced for the learned $\mathbf{A}$ and $\Delta$ parameters.

# 3   Knowledge Distillation for Aligned LMs

Knowledge distillation (KD) [35] serves as a compression technique aimed at training a smaller network that mimics the behavior of a larger teacher network. After initializing the model from the Transformer parameters, we aim to distill it to perform on par with the original language model. We assume that most of the knowledge from the transformer is maintained in the MLP layers which were transferred from the original model, and focus on distilling the fine-tuning and alignment steps of the LLM. During this stage, the MLP layers are kept frozen and the Mamba layers are trained as in Figure 1.

**Supervised Fine-Tuning** We first apply knowledge distillation to redo the supervised fine-tuning (SFT) stage of language model adaptation. During this stage, an LLM is trained to maximize the likelihood of a response $y$ given an input prompt $x$, i.e. $p(y \mid x)$. The task looks similar to conditional generation.

There are two common approaches for distillation in this setting. One method is to use word-level KL-Divergence. In this setting, the full probability distribution of the student model $p(\cdot; \theta)$ is trained to match the full distribution of the teacher model $p(\cdot; \theta_T)$ by minimizing the KL divergence over the entire set of next possible tokens at position $t$. The second method is sequence-level knowledge distillation (SeqKD) [39]. SeqKD suggests a simple method for distillation on this style of task, by replacing the ground truth text $y_{1\cdots t}$ with the teacher generation output $\hat{y}_{1\cdots t}$, also known as pseudo-labels.

$$\mathcal{L}(\theta) = -\sum_{t=1}^{T} \alpha \ \log \ p(\hat{y}_{t+1} \mid \hat{y}_{1:t}, x, \theta) + \beta \ \mathrm{KL}\left[p(\cdot \mid \hat{y}_{1:t}, x, \theta_T) \mid\mid p(\cdot \mid \hat{y}_{1:t}, x, \theta)\right] \quad (2)$$

Here $\theta$ is trainable parameters of the student model and $\alpha$ and $\beta$ control the weights of sequence and word loss term respectively.

**Preference Optimization** The second stage of instruction-tuning for LLMs is to align them to a set of user preferences. During this stage, a set of desired preference pairs is used to improve the model's output. The objective is to produce outputs $y$ to prompts $x$ that maximize a reward model $r$ while maintaining close to a reference model. Typically the reference model is chosen to be the model after supervised fine-tuning. For distillation, we can conveniently utilize the original teacher, i.e.

$$\max_{\theta} \mathbb{E}_{x\sim\mathcal{D}, y\sim p(y|x;\theta)}\left[r_\phi(x, y)\right] - \beta\mathrm{KL}\left[p(y \mid x; \theta) \mid\mid \pi(y \mid x; \theta_T)\right] \quad (3)$$

This preference model is defined by a reward function $r_\phi(x, y)$ dependent on the method used. Previous research utilizing AI feedback has primarily focused on employing reinforcement learning methods, such as proximal policy optimization (PPO) [64], to optimize $\phi$ concerning this reward. Recently, methods using direct preference optimization (DPO) [58] have been effective at optimizing this objective with direct gradient updates. Specifically, DPO shows that, if we have access to preferred $y_w$ and dispreferred $y_l$ outputs for a given prompt $x$, we can reformulate this optimization problem as,

$$\pi_\theta = \max_{\theta} \mathop{\mathbb{E}}_{(x,y_w,y_l)\sim\mathcal{D}} \log \sigma\left(\beta\log\frac{p(y_w|x;\theta)}{p(y_w|x;\theta_T)} - \beta\log\frac{p(y_l|x;\theta)}{p(y_l|x;\theta_T)}\right). \quad (4)$$

This optimization can be performed at the sequence level by scoring the preferred and dispreferred outputs of the model with the teacher and student and then backpropagating to the student. As far as we are aware this is the first use of DPO as a distillation objective.

# 4   Speculative Decoding Algorithms For Linear RNNs

The main goal of the linear RNN formulation is to improve decoding efficiency. For both attention and linear RNNs, the serial dependency of autoregressive generation inherently bottlenecks efficiency. Systems cannot utilize all available compute, as they need to wait for the generation of previous tokens to proceed [67, 41, 11, 76, 10]. *Speculative decoding* has emerged as a method for breaking this bottleneck by spending extra compute to speculate

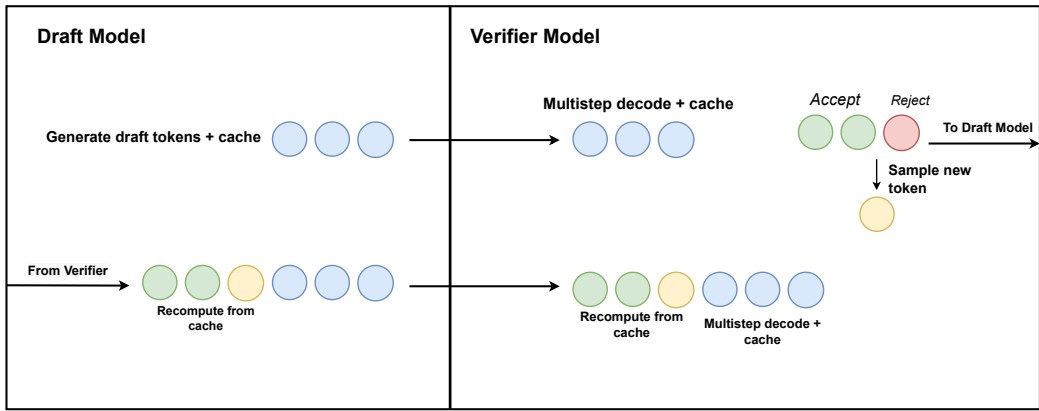

Figure 2: Multi-Step RNN Speculative Decoding. *Left (top)*: The draft model generates the set of blue draft tokens sequentially. The draft tokens are then verified. *Right (top)*: Verification uses the multistep kernel, without materializing the intermediate states. The last token is rejected and replaced with the true best token. Note, that even though more tokens are generated we cannot advance the hidden state cache. *Left (bottom)* The draft model can now generate more blue draft tokens from the current tokens, resulting in six total. *Right (bottom)* When the new draft is verified, the multi-step kernel returns both the hidden state after the yellow token and the final hidden state, since verification will fall between those positions. on future generations. In this section, we consider approaches for applying this technique to large Mamba models, which can then be applied to the distilled models.

## 4.1 Challenges in RNN Speculation

Speculative decoding uses two models: a draft model, $\theta_D$, and a verification model, $\theta_V$. The fast draft model produces potential future completions, $y^* = \arg\max_{y_{1:T}} p(y_1, \ldots, y_T; \theta_D)$, and the larger verification model checks that these are top ranking at each time step, i.e. checking $p(y_t^*|y_{1:t-1}^*; \theta_V)$. The longer a chain before a verification failure the faster the output. If a partial chain matches, we can rewind to the last match.

Attention-based models are particularly amenable to speculation, as they are slow at generation due to sequential nature, but fast at verification due to their ability to check multiple tokens in parallel. Linear RNN models like Mamba have significantly different performance characteristics that make them less amenable to speculative decoding. Sequential decoding using recurrent-style sampling is already significantly faster than attention. Like attention, there are parallel modes for models like Mamba which are used at training. These are efficient, but are tuned for extremely long sequences. In addition, they rely on hardware-aware optimizations, such as avoiding materializing intermediate states. These properties make it difficult to use for speculation for relatively short chains when it is unknown when a conflict will occur.

An additional challenge arises from caching states in RNN models. The state of an attention model is represented by the key-value cache, $\mathbf{K}_{1:t}, \mathbf{V}_{1:t}$; whereas the state of an RNN model is simply $\boldsymbol{h}_t$. To be competitive with attention this single RNN state needs to be very large. During speculation, we need to rewind to a previous state at time step $t'$. For attention, this is simply $\mathbf{K}_{1:t'}, \mathbf{V}_{1:t'}$; however, for RNNs this would require caching all $\boldsymbol{h}_{1:t}$ which would require a large memory overhead.

## 4.2 Multi-Step Linear RNN Speculation

We propose a new algorithm for linear RNN speculative decoding using hardware-aware multi-step generation. The core to the approach generation kernel that computes,

$$y_{j:k}, \boldsymbol{h}_j, \boldsymbol{h}_k \leftarrow \text{MultiStep}(\boldsymbol{h}_i, y_{1:n}, i, j, k; \mathbf{A}, \mathbf{B}, \mathbf{C}, \Delta)$$

Where $i$ is the starting hidden state, $i \leq j \leq k$, and $j \ldots k$ is the range of $\boldsymbol{y}$ outputs needed. The kernel is hardware-aware because it avoids materializing key terms off of the fast GPU

memory. Specifically, it avoids instantiating most $\boldsymbol{h}_{1:n}$ as well as the discrete-time linear RNN parameters. This kernel is aimed to target the issues presented above. Specifically, it can save a snapshot of the state $\boldsymbol{h}_j$ before evaluating the draft tokens. This allows recomputing the correct state on the fly after a token is rejected. The assumption is that decoding is bottlenecked by memory and not by compute, as we can compute multiple steps of decoding with very little overhead over single-step decoding.

Algorithm 2 and Figure 2 show the full algorithm. The approach maintains only one RNN hidden state in cache for verification and advances it lazily based on the success of the multi-step kernel. Since the distilled models contain transformer layers, we also extend speculative decoding to Attention/RNN hybrid architectures. In this setting, the RNN layers perform verification according to Algorithm 2, while the transformer layers simply perform parallel verification.

Note that if the draft model is a Mamba or hybrid model, the speculation part of the algorithm gets more complicated, as the draft model needs to recompute the state for the tokens accepted in the previous iteration. This is done similarly to the verifier model, by caching older entries and recomputing on the fly during the next round of speculation.

**Algorithm 2** Multi-Step Linear RNN Speculation

> **function** VERIFY($y_{1:k}, j, \boldsymbol{h}_i$)
> $\quad$ // $y_{1:k}$ are draft, $j$ is last verified,
> $\quad$ // $\boldsymbol{h}_i$ is a cached state with $i \leq j$
> $\quad y'_{j:k}, \boldsymbol{h}_j, \boldsymbol{h}_k \qquad\qquad\qquad \leftarrow$
> MULTISTEP($\boldsymbol{h}_i, y_{1:k}, i, j, k; \theta_v$)
> $\quad\quad k' \leftarrow$ FIRSTCONFLICT($y_{j:k}, y'_{j:k}$)
> $\quad\quad$ **return** $k', \boldsymbol{h}_k$ if $k' = k$ else $\boldsymbol{h}_j$
> **function** SPECULATE(K)
> $\quad$ // $K$ tokens are drafted per step
> $\quad \boldsymbol{h}_{\text{cache}} \leftarrow \boldsymbol{h}_0$
> $\quad j \leftarrow 0$
> $\quad$ **while** $y_j$ is not end **do**
> $\quad\quad k \leftarrow j + K$
> $\quad\quad y_{j+1:k} \leftarrow \arg\max p(y_{j+1:k} \mid y_{1:j}, \theta_D)$
> $\quad\quad j, \boldsymbol{h}_{\text{cache}} \leftarrow$ VERIFY($y_{1:k}, j, \boldsymbol{h}_{\text{cache}}$)
> $\quad$ **return** $y_{1:j}$

### 4.3 Speculation Analysis and Hardware Specific Optimization

To verify the effectiveness of this approach we run the speculation using Mamba 7B and Mamba 2.8B as target models. Results are shown in Table 1. Figure 3 shows the performance characteristics of the Multi-Step kernel itself.

| Model Size | GPU | K | # Gen. Tokens | Throughput (toks/s) | Speedup |
|---|---|---|---|---|---|
| 2.8B | 3090 | 3 | 3.01 | 259 | 2.3x |
| 2.8B | 3090 | 4 | 3.28 | 289 | 2.6x |
| 2.8B | H100 | 3 | 4.04 | 389 | 1.71x |
| 2.8B | H100 | 4 | 3.9 | 421 | 1.85x |
| 7B | 3090 | 3 | 3.19 | 109 | 2.1x |
| 7B | 3090 | 4 | 3.56 | 110 | 2.1x |
| 7B | H100 | 3 | 3.28 | 271 | 1.95x |
| 7B | H100 | 4 | 3.6 | 272 | 2x |

Table 1: Speedup results for speculative decoding with pure Mamba models. The 2.8B verifier uses a 130M Mamba draft. The 7B verifier uses a Llama3 1B draft we trained. Data is from The Pile. $K$ is number of draft tokens produced, # *Gen* includes an additional token from the last verifier logits.

**Speedup on H100 GPUs.** A naive implementation of our algorithm already shows strong performance on Ampere GPUs as shown in Table 1. However, achieving strong performance on H100 GPUs is much more challenging. This is mainly due to GEMM operations being much faster, which makes the overhead incurred from the caching and recomputation operations more visible. In practice, the naive implementation of our algorithm, with several different kernel

Figure 3: Performance of the multi-step SSM kernel for generating 32 tokens.

calls, achieves a decent speedup on 3090 GPUs (1.5x for Mamba 2.8B with $60\%$ acceptance rate) but no speedup at all on H100s.

We optimized our implementation by fusing kernels, and by adapting the implementation to easily allow caching and recomputing old steps. Specifically, the verifier model performs i) recomputation of previous steps from the cache, ii) multistep decoding for the new sequence of draft tokens and iii) caching within a single kernel [2]. For the draft model, recomputation, decoding and caching are also fused in a single kernel. The resulting implementations archives speedups on H100s GPUs, as shown in Table 1.

## 5 Results

### 5.1 Experimental Setup

**Target models.** We perform experiments using two LLM chat models: Zephyr-7B [72], which is a chat fine-tuned Mistral 7B [37], Llama-3 Instruct 8B [21]. For the linear RNN models, we use hybrid versions of Mamba and Mamba2 with 50%, 25%, 12.5%, and 0% attention layers. We refer to 0% as a pure Mamba model. Mamba2 is a variant architecture of Mamba that is designed to be more targeted to recent GPU architectures. Zephyr-Mamba refers to a distillation from Zephyr [72], while Llama3-Mamba / Llama3-Mamba2 indicates distillation from Llama-3 instruct 8B [71]. Strictly speaking, our distilled Mamba-Zephyr is a subquartic model, since Zephyr/Mistral-8B uses sliding window attention architecture. Our distilled Mamba-Zephyr (50%) has the similar architecture as Samba [60].

**Training.** Distillation does not require any language modeling pretraining data, but instead uses the post-training process to adapt the new model. We use a three-stage process. In the first stage, we use UltraChat [20] and UltraFeedback [17] as seed prompts and use the teacher model to generate pseudo-labels. The student model is trained in one epoch using the loss $\mathcal{L}$ in Eq 2 with $\alpha = 1$ and $\beta = 0.1$. Models are trained using AdamW optimizer with $\beta = (0.9, 0.98)$ with a batch size $64$. We use a linear learning rate warm-up (for the first $500$ steps) followed by cosine annealing. In the second stage, we use supervised finetuning with our model on the GenQA [12], InfinityInstruct [3] and OpenHermes 2.5 [70] datasets using SFT in one epoch, with the same hyperparameters as Zephyr [72]. In the final stage, for models distilled from Zephyr, we do distilled alignment with our model using DPO on the UltraFeedback [17] dataset which is consistent with teacher model. While models distilled from Llama-3 instructed 8B, we use datasets from SimPO [51] and Zephyr [72]. We only freeze Gated MLP (FFN) in the first stage, while in the second and final stage all parameters are trained [3]. The total distillation process for each hybrid model (e.g., Mamba-Llama3 (50% att)) takes less than five days in 8x80G A100.

**Baselines.** In addition to the core Transformer architectures, the main baselines we compare against are other large-scale linear RNN models. We compare with both pure SSM architectures, such as TRI Mamba 7B [52] trained with 1.2T tokens and Falcon Mamba 7B[4] trained with more than 5T tokens, hybrid SSM architectures, such as Nvidia Hybrid Mamba 2 [74] trained with 3.7T tokens, and other linear hybrid RNN models, such as Recurrent Gemma-9B Instruct [8, 19].

After the release of the new SoTA transformer models at the 8B and 3B scales, Llama-3.1 and Llama-3.2, we have streamlined the distillation process and are now distilling using the larger Llama-3.1 70B teacher model while initializing models with similarly sized 3B and 8B scales, respectively. We distill our model on the GenQA [12] and InfinityInstruct [3] datasets, resulting in Mamba-Llama3.2-3B, Mamba2-Llama3.2-3B, Mamba-Llama3.1-8B, and Mamba2-Llama3.1-8B. Additionally, we perform further DPO on top of these models using the same dataset as before, resulting in Mamba-Llama3.2-3B-dpo, Mamba2-Llama3.2-3B-dpo, Mamba-Llama3.1-8B-dpo, and Mamba2-Llama3.1-8B-dpo. The distillation phase takes eight days on 8xA100 and four days on 8xH100.

---

[2]Additionally, we implement the convolutional part of the Mamba block using a circular buffer which allows us to keep track of the old entries and include them in the convolution when they are needed for recomputation.

[3]We freeze the MLP layers in the first stage because we want to produce a model similar to the initialization model. However, in the end-to-end distillation, we only focus on the KL loss, so training all parameters (not freezing the MLP layers) will give better results.

[4]https://huggingface.co/tiiuae/falcon-mamba-7b

## 5.2 Evaluation on Chat Benchmarks

We evaluate our models using both single-turn, AlpacaEval [43] and multi-turn chat benchmarks, MT-Bench [84]. These benchmarks assess the model's ability to follow instructions and respond to challenging prompts across a wide variety of domains.

| Model (% Att) | Size | Align | MT-Bench (score) | MT-Bench (Round 1) | MT-Bench (Round 2) | AlpacaEval (LC win %) | AlpacaEval (win %) |
|---|---|---|---|---|---|---|---|
| Zephyr | 7B | DPO | **7.34** | - | - | $13.20_{0.96}$ | $10.99_{0.96}$ |
| **Mamba-Zephyr (50%)** | 7B | DPO | 7.31 | - | - | $\mathbf{20.66_{0.74}}$ | $\mathbf{16.69_{1.10}}$ |
| **Mamba-Zephyr (25%)** | 7B | DPO | 7.03 | - | - | $17.16_{0.69}$ | $13.11_{1.00}$ |
| **Mamba-Zephyr (12.5%)** | 7B | DPO | 6.40 | - | - | $15.32_{0.66}$ | $12.96_{1.02}$ |
| Llama-3.1-Instruct | 8B | RLHF | **8.0** | - | - | **20.9** | **21.8** |
| **Mamba-Llama3.1 (50%)** | 8B | | 7.7 | 8.0 | 7.3 | $18.97_{1.23}$ | $21.22_{1.23}$ |
| **Mamba2-Llama3.1 (50%)** | 8B | | 7.6 | 8.1 | 7.0 | $18.99_{1.24}$ | $21.55_{1.24}$ |
| **Mamba-Llama3.2 (50%)** | 3B | | 6.9 | 7.6 | 6.1 | $13.57_{1.08}$ | $15.54_{1.08}$ |
| **Mamba2-Llama3.2 (50%)** | 3B | | 6.5 | 7.1 | 5.8 | $12.61_{1.05}$ | $14.34_{1.05}$ |
| Llama-3-Instruct | 8B | RLHF | **8.00** | - | - | $22.90_{1.26}$ | $22.60_{1.26}$ |
| **Mamba-Llama3 (50%)** | 8B | DPO | 7.35 | 7.82 | 6.88 | $\mathbf{29.61_{1.31}}$ | $\mathbf{26.69_{1.31}}$ |
| **Mamba-Llama3 (25%)** | 8B | DPO | 6.86 | 7.56 | 6.15 | $25.85_{1.26}$ | $22.50_{1.26}$ |
| **Mamba-Llama3 (12.5%)** | 8B | DPO | 6.46 | 6.91 | 6.01 | $20.76_{1.16}$ | $17.93_{1.16}$ |
| **Mamba2-Llama3 (50%)** | 8B | DPO | 7.32 | 7.93 | 6.70 | $26.78_{1.26}$ | $22.69_{1.26}$ |
| **Mamba2-Llama3 (25%)** | 8B | DPO | 6.74 | 7.24 | 6.24 | $22.75_{1.18}$ | $19.01_{1.18}$ |
| **Mamba2-Llama3 (12.5%)** | 8B | DPO | 6.48 | 6.83 | 6.13 | $20.25_{1.13}$ | $16.88_{1.13}$ |
| **Mamba2-Llama3 (0%)** | 8B | DPO | 5.64 | 6.16 | 5.11 | $14.49_{0.93}$ | $10.88_{0.93}$ |
| Falcon Mamba Instruct | 7B | SFT | 6.40 | 7.25 | 5.55 | $4.04_{0.45}$ | $2.15_{0.45}$ |
| GPT-3.5-turbo | - | RLHF | 7.94 | - | - | 22.70 | 14.10 |
| GPT-4o | - | RLHF | - | - | - | $\mathbf{57.46_{1.47}}$ | $\mathbf{51.33_{1.47}}$ |

Table 2: Chat benchmark results for open-access and proprietary models on MT-Bench and AlpacaEval. MT-Bench scores model responses using GPT-4. AlpacaEval version two measures the win-loss rate between baseline models and GPT-4 scored by GPT-4 Turbo.

Table 2 shows the performance of our models on chat benchmarks compared with large transformer models. The distilled hybrid Mamba model (50%) achieves a similar score in the MT-benchmark as the teacher model, and slightly better than the teacher model on the AlpacaEval benchmark in both LC win rate and overall win rate. The distilled hybrid Mamba (25% and 12.5%) performance is slightly worse than that of the teacher models in the MT benchmark but still surpasses some large transformers even with more parameters in AlpacaEval. The distilled pure (0%) model does degrade significantly in accuracy. Notably, the distilled hybrid model performs better than Falcon Mamba, which was trained from scratch with more than 5T tokens.

## 5.3 Evaluation on General Benchmarks

**Zero Shot Evaluation.** We utilize the open-source LM Evaluation Harness library [25] (branch `big-refactor`) to assess 10 tasks, with the following evaluation metrics: WinoGrande (WG) accuracy [62], PIQA (PQ) accuracy [7], HellaSwag (HS) normalized accuracy [82], ARC-Easy and ARC-Challenge (AE and AC) accuracy and normalized accuracy, [15], MMLU (MM), accuracy [33], OpenBookQA (OB) normalized accuracy [54], TruthFulQA (TQ) accuracy [46], PubMedQA (PM) accuracy [38], and RACE (RA), accuracy [40]. Each task is evaluated by analyzing the probability assigned by the model to each potential answer choice.

Table 3 shows zero shot evaluation in LM Eval benchmark for Mamba and Mamba2 distilled from different teacher models. Both hybrid Mamba-Llama3 and Mamba2-Llama3 models, distilled from the Llama-3 Instruct 8B, perform better compared to the open-source TRI Mamba and Nvidia Mamba models trained from scratch. Performance degrades with more linear RNN layers, but is still competitive at 25% to models trained from scratch.

## 5.4 Hybrid speculative decoding

**Setup** We perform speculative decoding using the distilled hybrid models. We run experiments using both Hybrid Mamba 50% and Hybrid Mamba 25% as main models. For the draft models, we train 2 and 4-layer Transformer Draft models on the OpenHermes2.5

| Model (% Att) | WG | PI | HS | AE | AC | MM | OB | TQ | PM | RA | AVG |
|---|---|---|---|---|---|---|---|---|---|---|---|
| TRI Mamba-7B | 71.42 | 81.01 | 77.93 | 77.53 | 46.67 | 33.39 | **46.20** | 32.09 | 72.30 | 37.99 | 57.65 |
| Nvidia Hybrid Mamba-8B | 71.27 | 79.65 | 77.68 | 77.23 | 47.70 | 51.46 | 42.80 | 38.72 | 69.80 | 39.71 | 59.60 |
| Llama-3.1-8B-Instruct | 73.88 | **80.79** | 79.21 | 81.78 | 55.20 | **68.12** | 43.20 | 42.67 | **75.20** | 44.78 | 64.48 |
| **Llama3.1-Mamba (50%)** | 72.77 | 79.33 | 75.91 | 82.24 | 53.84 | 62.13 | 42.80 | 40.02 | 72.00 | 42.11 | 62.32 |
| **Llama3.1-Mamba-DPO (50%)** | 73.80 | 80.41 | 77.36 | 84.01 | 56.57 | 63.50 | 44.20 | 46.07 | 74.40 | 43.44 | 64.38 |
| **Llama3.1-Mamba2 (50%)** | 71.74 | 78.89 | 75.36 | 82.20 | 52.65 | 61.01 | 41.60 | 40.31 | 72.60 | 42.11 | 61.85 |
| **Llama3.1-Mamba2-DPO (50%)** | **74.11** | 80.03 | **79.69** | **84.81** | **59.73** | 59.74 | 44.00 | 50.22 | 74.60 | **46.12** | **65.31** |
| Llama-3.2-3B-Instruct | 67.48 | 75.68 | 70.43 | 74.07 | 45.90 | **60.43** | 36.00 | 38.01 | 69.60 | 40.67 | 57.83 |
| **Llama3.2-Mamba (50%)** | 67.32 | 77.31 | 70.37 | 77.65 | 48.38 | 54.48 | 39.40 | 42.02 | 66.40 | 40.29 | 58.36 |
| **Llama3.2-Mamba-DPO (50%)** | 67.40 | 77.31 | 72.56 | 79.97 | 52.65 | 55.09 | 41.60 | 48.53 | **70.00** | **43.64** | **60.88** |
| **Llama3.2-Mamba2 (50%)** | 66.06 | 76.01 | 69.13 | 76.68 | 46.67 | 53.12 | 38.80 | 34.78 | 63.80 | 39.81 | 56.49 |
| **Llama3.2-Mamba2-DPO (50%)** | 67.32 | **77.69** | 74.45 | 80.26 | 54.10 | 52.47 | 42.40 | 50.28 | 65.40 | 43.44 | 60.78 |
| **Mamba-Zephyr (50%)** | 68.82 | 80.36 | 76.91 | 81.40 | 55.63 | 55.43 | 42.60 | 41.99 | 72.60 | 42.20 | 61.79 |
| **Mamba-Llama3 (50%)** | 68.98 | 78.02 | 78.43 | 74.45 | 51.96 | **57.81** | 44.00 | 47.69 | 73.00 | 38.56 | 61.30 |
| **Mamba-Llama3 (25%)** | 62.83 | 78.07 | 75.00 | 74.28 | 47.35 | 53.50 | 40.00 | 43.64 | 65.40 | 36.94 | 57.70 |
| **Mamba-Llama3 (12.5%)** | 59.75 | 75.08 | 71.71 | 70.58 | 43.60 | 49.81 | 41.40 | 41.41 | 62.40 | 34.45 | 55.02 |
| **Mamba2-Llama3 (50%)** | **71.51** | **81.45** | **79.47** | 78.83 | 58.19 | 55.70 | 44.20 | **57.74** | 72.4 | 38.85 | **63.84** |
| **Mamba2-Llama3 (25%)** | 64.80 | 78.73 | 77.7 | 76.35 | 52.47 | 53.71 | 42.40 | 55.33 | 64.80 | 39.23 | 60.55 |
| **Mamba2-Llama3 (12.5%)** | 63.38 | 76.82 | 73.14 | 75.84 | 50.26 | 50.78 | 39.60 | 50.00 | 65.80 | 36.46 | 58.21 |
| **Mamba2-Llama3 (0%)** | 58.56 | 76.82 | 70.75 | 74.12 | 47.95 | 45.19 | 39.00 | 40.20 | 62.20 | 32.63 | 54.74 |

Table 3: Evaluation on LM Eval benchmark for Mamba and Mamba2 distilled from Llama-3 Instruct 8B.

dataset [70], for approximately 3 full epochs, following the "shrink and fine-tune" approach from [66]. Specifically, we initialize the draft layers using layers from the Zephyr-7B model (we take layers at indices $[0, 31]$ for the 2-layer model and $[0, 10, 20, 31]$ for the 4-layer model), and the embeddings and language model head also from the Zephyr-7B model [72]. We perform loss masking on the prompt, thus only considering next token prediction loss (cross-entropy) on the chat continuations from the training set. Speculative decoding experiments are run on a single NVIDIA RTX 3090 on data from OpenHermes2.5.

| Draft Model | K | Target Model (% Att) | # Gen. Tokens | Speedup |
|---|---|---|---|---|
| 2 layers | 4 | Mamba-Zephyr (50%) | 2.48 | 1.8x |
| | 4 | Mamba-Zephyr (25%) | 2.64 | 1.88x |
| 4 layers | 4 | Mamba-Zephyr (50%) | 3 | 1.81x |
| | 4 | Mamba-Zephyr (25%) | 3 | 1.8x |
| 4 layers | 3 | Mamba-Llama3 (50%) | 2.7 | 1.6x |
| 4 layers | 4 | Mamba-Llama3 (50%) | 3.6 | 1.58x |

Table 4: Performance metrics for different draft and target model configurations for $K = 4$ on data from OpenHermes2.5. # *Gen* is the average number of generated tokens per speculative decoding step and includes an additional token from the last verifier logits.

**Results**    Table 4 shows results for hybrid speculative decoding with, using both the Zephyr and Llama hybrid models with different configurations. For both the $50\%$ and $25\%$ distilled models, we achieve speedups of over $1.8$x on the Zephyr-Hybrid compared to the non-speculative baseline. We also show that the 4-layer draft model we trained achieves a higher acceptance rate, but it adds some additional overhead due to the increased draft model size. For the Llama-hybrid models, the speedups are more modest since the draft model is larger due to the large embedding table of Llama 3. In subsequent work, we will focus on making these draft models smaller.

## 6   Conclusion

We consider the problem of maintaining LLM abilities while increasing decoding speed through a combination of distillation and speculative decoding. We first show that a Transformer LLM can be used to effectively initialize a Mamba linear RNN model while maintaining original abilities. We then show that through a combination of distillation on supervised instructions and preferences, we can improve the model's ability with relatively little compute. Finally, we show that the Mamba model can be significantly sped up at inference time through the use of a hardware-aware speculative decoding method. The full model nears LLM chat accuracy, and is accelerated with speculative decoding. We believe these results show that transformer knowledge can be transferred effectively to other architectures, opening up the potential for customizing the inference profile of LLMs beyond optimizing attention.

**Acknowledgement**

We thank Together AI for providing compute for some of the experiments. This work has benefited from helpful discussions with Albert Gu at CMU, François Fleuret and Vincent Micheli at the University of Geneva, Albert Tseng and Wen-Ding Li at Cornell University.

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

# A   Evaluation on Long Context Tasks

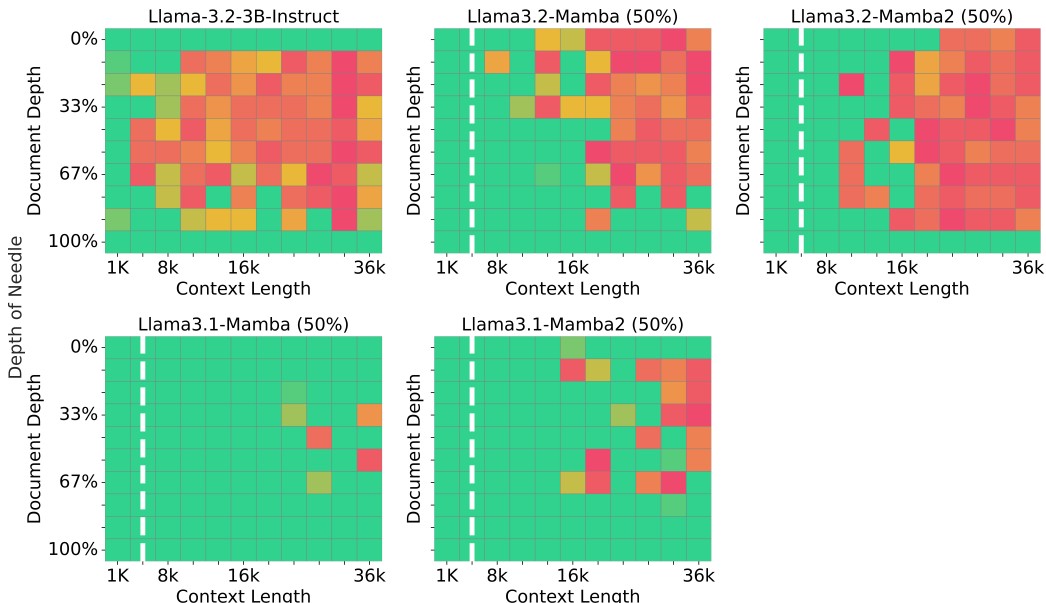

Figure 4: Needle in a Haystack evaluation. Green squares represent a high retrieval success rate, while the white dashed line marks the longest examples encountered during distillation training. The Y-axis indicates the distance to the retrieved target.

Figure 4 illustrates the results of Needle in a Haystack. Although the distillation length is only 2k, our distilled 3B models (Mamba-Llama3.2-3B (50%) and Mamba2-Llama3.2-3B (50%)) achieve perfect accuracy up to 10k, which is better than Llama-3.2-3B-Instruct. Similarly, the distilled 8B models (Mamba-Llama3.1-8B (50%) and Mamba2-Llama3.1-8B (50%)) achieve perfect accuracy up to 16k, with Mamba-Llama3.1-8B demonstrating good results up to 38k.

# B   Benchmark Evaluation

We also report few-shot evaluations on OpenLLMLeaderboard by conducting 25 shots on ARC-Challenge [14], 10 shots on HellaSwag [82], 5 shots on MMLU [34], and 5 shots on Winogrande [62]. For TruthFulQA, the mc2 metric is reported in this benchmark. For GSM8K [16], we follow the evaluation for instruct tuned model [51], which uses ZeroEval [45], a benchmark designed for chat models. We also include the CRUX [29] from that benchmark, which is designed for evaluating reasoning on code. All models are evaluated with greedy decoding in the ZeroEval.

Table 5 shows that the performance of our distilled hybrid models matches that of the best open-source linear RNN models on the Open LLM Leaderboard, while outperforming their corresponding open-source instruct models in GSM8K and CRUX.

# C   Analysis

**Comparison with other distillation approaches**   Table 6 (left) compares the perplexity of different model variants. We distill using Ultrachat as seed prompt [20] in one epoch and compare the perplexity. We find that removing more layers gets significantly worse. We also compare our distillation approach with a previous baseline. This approach distills a Transformer model into a Hyena model [57], as proposed in [59]. They use a different distillation approach using progressive knowledge transfer, wherein the student model is trained starting from the first layer and progressively extending to subsequent layers. While

| Model (% Att) | ARC | HS | MMLU | WG | TQ | GSM8K | CRUX |
|---|---|---|---|---|---|---|---|
| Falcon Mamba-7B | 62.03 | 80.82 | 62.11 | 73.64 | 53.42 | 41.32 | 8.88 |
| RecurrentGemma-9B | 52.00 | 80.40 | 60.50 | 73.60 | 38.60 | 38.51 | 26.25 |
| **Mamba-Llama3 (50%)** | 56.57 | 78.99 | 59.26 | 69.06 | 58.85 | 67.85 | 27.88 |
| **Mamba-Llama3 (25%)** | 55.03 | 75.66 | 52.68 | 62.83 | 55.03 | 40.64 | 15.62 |
| **Mamba-Llama3 (12.5%)** | 52.90 | 72.46 | 49.20 | 59.19 | 53.00 | 26.91 | 11.25 |
| **Mamba2-Llama3 (50%)** | 60.41 | 77.97 | 56.67 | 71.35 | 66.60 | 59.36 | 24.88 |
| **Mamba2-Llama3 (25%)** | 59.22 | 76.88 | 53.94 | 64.88 | 64.64 | 38.13 | 13.25 |
| **Mamba2-Llama3 (12.5%)** | 53.33 | 72.16 | 50.85 | 63.61 | 61.12 | 35.03 | 10.25 |
| **Mamba2-Llama3 (0%)** | 53.51 | 70.31 | 44.21 | 58.91 | 52.31 | - | - |

Table 5: Results on the Open LLM Leaderboard and ZeroEval Leaderboard. For GSM8K and CRUX, we chose the zero-shot evaluation using ZeroEval, which is designed for evaluating instruct models. We evaluated the corresponding instruct-tuned models for Falcon Mamba-7b and RecurrentGemma-9B, specifically Falcon Mamba-7b-instruct and RecurrentGemma-9B-it.

| Model (% Att) | PPL | Ratio |
|---|---|---|
| Teacher: Zephyr (7B) | 2.02 | 1 |
| Mamba-Zephyr (50%) | 2.09 | 1.03 |
| Mamba-Zephyr (25%) | 2.20 | 1.09 |
| Mamba-Zephyr (6.25%) | 2.46 | 1.22 |
| Mamba-Zephyr (0%) | 3.36 | 1.66 |
| Teacher: Pythia (70M) | 51.4 | 1 |
| Distill Hyena | **121.2** | 2.36 |

| Model | Hyb Mamba (50% Att) | Hyb Mamba (25% Att) |
|---|---|---|
| Dis | 5.55 | 5.01 |
| Dis+SFT | 5.61 | 4.97 |
| Dis+DPO | 5.42 | 4.84 |
| Dis+SFT+DPO | **6.69** | **6.10** |

Table 6: (Left) Perplexity comparison between our distillation approach and [59]. (Right) Ablation study of different alignment methods of the Distilled Hybrid Mamba on the MT-benchmark using OpenHermes 2.5 as the SFT dataset.

it is challenging to compare, our distill shows a smaller degradation (1.03 for 50 % attention, 1.09 for 25 % attention, 1.22 for 6.35% attention, and 3.36 for no attention), while the Distill Hyena model is trained in WikiText [53] dataset with a much smaller model and shows large perplexity degrade.

**Does distilling from preferences help?** In Table 6 (Right), we show the impact of different steps in the alignment process of the distillation. We observe that SFT or DPO alone does not yield much improvement, while SFT + DPO yields the best score. Models are trained using Zephyr as the teacher model and the OpenHermes 2.5 [70] dataset as the SFT dataset, and UltraFeedback [17] as the DPO dataset.

**Pseudo Label Distillation Ablations.** We consider several different model ablation studies in Table 7. For these experiments we consider training for 5k steps using the pseudo-label approaches on the Ultrachat [20] dataset. Table 7 (Left) presents the results of distillation with various initializations. According to this table, initializing weights from a transformer

| Model | Mamba (0% Att) | | Hyb Mamba (50% Att) | |
|---|---|---|---|---|
| | Froz | -Froz | Froz | -Froz |
| + Attention-Init | **3.36** | 66.7 | **2.09** | 9.1 |
| -Attention-Init | 18.2 | 20.3 | 7.4 | 11.2 |

| Model | Hyb Mamba (25% Att) | | Hyb Mamba (50% Att) | |
|---|---|---|---|---|
| | Step | -Step | Step | -Step |
| + Interleave | **2.20** | 2.29 | **2.09** | - |
| -Interleave | 2.89 | - | 2.41 | - |

Table 7: (Left) Perplexity comparison with different initialization at first stage. (Right) Perplexity comparison with different Mamba interleaving layers and stepwise distillation at first stage.

is crucial for performance. Without weight initialization from a transformer, perplexity significantly worsens for both pure Mamba models and hybrid models. Also, freezing MLP layers can help the student model focus on learning the interaction of tokens and better mimic attention layers. Table 7 (Right) shows also see smaller benefits from progressive distillation and interleaving the attention layers with Mamba.

**Attention Initialization.** We compare the default random initialization of Mamba with reusing the linear projection from the attention using the same recipe. Both models are trained using Zephyr as the teacher model and the OpenHermes 2.5 [70] dataset as the SFT dataset, and UltraFeedback [17] as the DPO dataset.

| Model | LAMBADA (ppl) | MMLU | ARC-C | TruthfulQA | HellaSwag | MT-Bench (score) | AlpacaEval (LC win %) |
|---|---|---|---|---|---|---|---|
| + Attention init | 6.20 | **47.98** | **49.15** | **46.67** | **75.07** | **6.69** | **14.11** |
| - Attention init | 55.01 | 26.21 | 25.26 | 34.01 | 27.91 | 1.04 | 0.02 |

Table 8: Performance of Zephyr-Mamba (50% attention) with different initialization.

Table 8 compares the performance of the hybrid model using two different initialization methods: default random initialization and reusing the linear projection from the attention. The model performs significantly better with reusing the linear projection from the attention compared to random initialization, across all evaluated benchmarks. This result confirms that initialization from attention weights is critical.

| Model | LAMBADA (ppl) | MMLU | ARC-C | TruthfulQA | HellaSwag | MT-Bench (score) | AlpacaEval (LC win %) |
|---|---|---|---|---|---|---|---|
| **50% Att w Mamba** | 6.20 | **47.98** | **49.15** | **46.67** | **75.07** | **6.69** | **14.11** |
| 50% Att w/o Mamba | 151.98 | 24.46 | 21.93 | 32.39 | 27.91 | 1.01 | 0 |

Table 9: Performance of Hybrid-Mamba with different initialization.

**Necessity of Linear RNN.** We train a model that removes Mamba blocks from the model entirely using the same recipe to see if the model can adapt. Both models are trained using Zephyr as the teacher model, with the OpenHermes 2.5 [70] dataset as the SFT dataset and UltraFeedback [17] as the DPO dataset. Table 9 compares the performance of the model with and without Mamba blocks. The model with Mamba performs significantly better than the one without it. This confirms that adding Mamba layers is critical and that the improved performance is not solely attributable to the remaining attention mechanism.

## D  Related Work

**Attention-free models.** Attention-free models offer improved computational and memory efficiency, making them increasingly popular for various language processing tasks, including autoregressive language modeling. Models like S4 [27] and its subsequent variants [31, 28] have shown promising results in long-range synthetic tasks [69]. Gated SSM architectures, such as GSS [50] and BiGS [75], incorporate a gating mechanism into SSMs for (bidirectional) language modeling. The recently introduced Mamba model [26] argues that the static dynamics of these methods fail to incorporate input-specific context selection within the hidden state, which could be crucial for tasks like language modeling. Mamba has been shown to outperform Transformers across different model sizes and scales. Additionally, several other sub-quadratic model architectures [57, 79, 19, 1, 2, 22, 4, 80] and hybrid architectures [23, 44] have also been proposed.

**Distillation from Transformers.** There have been relatively few attempts to distill on to linear RNN style models. Laughing Hyena [49] proposes to distill the long convolution into a state space representation, enabling constant time inference in Hyena [57]. Ralambomihanta et al. [59] introduces a progressive knowledge approach to distill small transformer models (70M) into Hyena models. [6]

**Speculative Decoding.** Speculative decoding [67, 41, 11, 76, 10] has recently emerged as a promising method to accelerate the inference process of large language models, particularly

Transformers. This approach utilizes a smaller draft model to speculatively generate candidate tokens, which the larger target model then verifies. Leviathan et al. [41], Chen et al. [11] proposed a rejection sampling scheme to improve inference quality, while Spector and Re [67] organized candidate tokens into a tree structure to enable more efficient verification. Subsequent work has examined both trained draft models [5, 13, 48] and training-free draft models [32, 78, 24].

