# OpenReview forum: "The Mamba in the Llama: Distilling and Accelerating Hybrid Models"
_NeurIPS.cc/2024/Conference — NeurIPS 2024 poster_

### Official Review · Reviewer_J8xQ · 2024-07-12

**Soundness:** 2
**Presentation:** 1
**Contribution:** 2
**Rating:** 4
**Confidence:** 5

**Summary:**

The paper focuses on distilling transformers into SSMs for fast inference after training. Specifically, the authors initialize states of SSMs by pretrained transformer components and utilize speculative decoding for multi-step predictions. Experiments show that the distilled hybrid model can achieve competitive performance compared to the teacher network, i.e. Zephyr-7B on standard Chat benchmarks after training with 3B tokens only. The distilled hybrid model can outperform Mamba 7B which is trained from scratch with 1.3T tokens.

**Strengths:**

How to accelerate the training process of Mamba-based hybrid models is a good research problem. In the paper, the authors propose to utilize distillation from pretrained transformer-based models. The results show the Mamba-based hybrid model can learn efficiently with fewer iterations and samples.

**Weaknesses:**

1. The writing is not well-organized and involves a lot of unclear parts. In Section 2.1, the derivations from softmax-based attention to linear RNNs (Mamba) are not clearly stated, e.g. unexplained symbols in equations. Also the derivations should obey some assumptions. Please refer to Mamba-2.
2. Some settings of the proposed method are not clear. For example, in Section 2.2, Why is A set as fixed over time? In Section 2.3, which components in the model are from attention and which parts are trainable. Figure 1 is also not informative without explanations of the colors and arrows. In section 4, what are the differences between speculative decoding used in transformer and that used in Mamba here? What are the details of hare-aware designs?
3. The experiments are not enough. For comparison methods, Hybrid Mamba which is trained from scratch should be included. Besides Mamba, other SSMs and linear RNNs should be included. The benchmarks are only on chat and academic tasks.

**Questions:**

see the weaknesses.

**Limitations:**

The paper should be generalized to all SSM models (including Mamba-1/2) or linear RNN models, instead of only focusing on Mamba-1.

---

> ### Author Rebuttal · Authors · 2024-08-07
>
> Thanks for your comments! Please find our responses below.
>
> $\mathcal{Q}.$ In Section 2.2, Why is A set as fixed over time?
>
> $\mathcal{A}.$ The equation after line 92 has a fixed-over-time $A$ because it is the continuous SSM, which is consistent with the Mamba paper. However, after discretization, we obtain a matrix $\bar{A}$ which depends on the timestamp $t$ since $\Delta$ is not fixed over time.
>
> $\mathcal{Q}.$ In Section 2.3, which components in the model are from attention and which parts are trainable. Figure 1 is also not informative without explanations of the colors and arrows.
>
> $\mathcal{A}.$
> In the caption of Figure 1, we state that we freeze the FFN layer and only train the projects and Mamba block. We also explain that weights in the same color are initialized from the same color in attention. Please check our updated one-page PDF for an updated explanation. We will clarify this in the final version.
>
> $\mathcal{Q}.$ In section 4, what are the differences between speculative decoding used in transformer and that used in Mamba here? What are the details of hare-aware designs?
>
> $\mathcal{A}.$ Speculative decoding requires verfying multiple tokens in parallel and rolling back when some of them are rejected. This is easily done in transformers by adjusting the size of the kv-cache. For SSMs and Mamba in particular, rolling back requires "undoing" the decoding and getting back to a previous state. A naive way of achieving this would be to cache the states and restore a previous state after tokens are rejected. In practice, this is very inefficent as it requires materializing several very large states.
> The solution we implement, which we summarize in Algorithm 2 in the paper, allows us to avoid caching the states, and instead recompute the states for the accepted tokens on-the-fly using the cached $x$, $B$ and $\Delta_t$ matrices, without ever materializing the intermediate states.
>
> $\mathcal{Q}.$ For comparison methods, Hybrid Mamba which is trained from scratch should be included.
>
> $\mathcal{A}.$ Please refer to the 2 and 3a for comparisons with the best Hybrid Mamba which is trained from scratch.
>
> $\mathcal{Q}.$ The benchmarks are only on chat and academic tasks.
>
> $\mathcal{A}.$ Please refer to section 2 in the general response. It includes 10 probability-based tasks in LM Bench, which are the same as Nvidia Hybrid Mamba.
>
> $\mathcal{Q}.$ In Section 2.1, the derivations from softmax-based attention to linear RNNs (Mamba) are not clearly stated, e.g. unexplained symbols in equations. Also the derivations should obey some assumptions.
>
> $\mathcal{A}.$ We apologize for the lack of clarity here. We were not trying to claim an equivalence between these equations, but instead trying to show the relationship. Please check our updated one-page PDF for an updated explanation. We will clarify this in the final version.
>
> $\mathcal{Q}.$ Please refer to Mamba-2.
>  Why did we choose Mamba-1 instead of Mamba-2?
>
> $\mathcal{A}.$ Mamba 2 was released on May 31, which is after the NeurIPS submission deadline. According to NeurIPS guideline, papers appearing less than two months before the submission deadline are generally considered concurrent to NeurIPS submissions.
>
> To show the generalization ability of our paper, we adopted our approach in Mamba 2 and report the numbers in section 2 in the general response. We would like to state that the goal of this project is to distill a large transformer into Mamba and generate fast. Indeed, our approach works for any linear RNN model and our story does not change using different linear RNN models. We hope to try out more linear RNN models as they are released.

---

> > ### Comment · Reviewer_J8xQ · 2024-08-13
> >
> > Thanks for the effects. I appreciate the mamba2 experiments provided, but I am still doubt the generalization of the proposed method to other models like linear RNN or linear attention. The paper should not only focus on mamba. I would like to raise my score to 4.

---

> > > ### Author Response · Authors · 2024-08-13
> > >
> > > We thank the reviewer for raising his score. However, we wonder why the reviewer believes we shouldn't focus on Mamba, since it's clear from the title that this is the goal and scope of our work. We would like to ask the reviewer to reconsider their score of rejection, unless there are other doubts or issues about our work, which we are happy to engage with.

---

> ### Comment · Reviewer_J8xQ · 2024-08-13
>
> My main concern is the narrow scope of the methods and experiments of this paper. The paper’s story seems to rely on a biased assumption that Mamba will replace transformers—an appealing idea, but speculative at this stage. The focus on distilling and accelerating the Mamba hybrid model, while potentially applicable to other models (like linear RNN or linear attention), still requires further demonstration. And the paper does not propose any new methods or theory, it is just a application of proven techniques on mamba. Given the limited scope and novelty, I don’t believe the paper meets the NeurIPS standard in its current form.

---

> ### Author Response · Authors · 2024-08-13
>
> We are very grateful for all the encouraging and constructive reviews. Please find our responses below.
>
> $\mathcal{Q}.$ The paper’s story seems to rely on a biased assumption that Mamba will replace transformers—an appealing idea, but speculative at this stage.
>
> $\mathcal{A}.$ We appreciate the reviewer's concerns. We do not claim in the paper that Mamba will replace transformers and in fact, believe that people will continue mainly training transformers. We instead think that some organizations may want prioritize fast and lower-memory inference, and therefore want to distill their transformer models to Mamba for some applications.
>
> $\mathcal{Q}.$ Additionally, the paper does not introduce any new method or theory.
>
> $\mathcal{A}.$ We think there are two new methods introduced by this paper. 1) Our distillation approach is novel. To the best of our knowledge, this is the first work demonstrating that transformations between large transformers and linear RNN models is possible through a mathematical transformation. And in our experiments, we distill different large transformers to support this claim.  2) We introduce a new algorithm for speculative decoding in the Mamba model which is efficient in practice. We agree that there is no new theory introduced in this work, although that is generally true for most deep learning papers.
>
> $\mathcal{Q}.$ We don't understand that it is just an application of proven techniques on mamba.
>
> $\mathcal{A}.$ While distillation is a proven technique, as far as we know there is no work published on distilling from transformer to Mamba or related linear RNNs. The details of what distillation approaches work and how they can reuse parameters is a the contribution. We are a bit perplexed by this comment, and it would be great help if you can point us any thing related to this. (Previous work on this topic does not do anything like distillation.)

---

### Official Review · Reviewer_pZLW · 2024-07-14

**Soundness:** 3
**Presentation:** 3
**Contribution:** 3
**Rating:** 5
**Confidence:** 4

**Summary:**

The paper presents a mechanism to distill from llama to mamba.

**Strengths:**

(1) Good paper writing with clear figures.
(2) The method is easy to understand.
(3) Good experiment design on speculative decoding and other distillation strategy.
(4) Good performance (Table 1, 2)

**Weaknesses:**

The experiments are a bit limited, only performed on chat corpora. Further data mixture is needed to understand the effectiveness of the approach. (Yet due to other strength, the reviewer thinks it should receive a positive score).

**Questions:**

Please see the weakness section.

**Limitations:**

Please see the weakness section.

---

> ### Author Rebuttal · Authors · 2024-08-07
>
> We sincerely appreciate your feedback. Please see our responses below.
>
> $\mathcal{Q}.$ The experiments are a bit limited, only performed on chat corpora. Further data mixture is needed to understand the effectiveness of the approach
>
> $\mathcal{A}.$ Please refer to 2 in general response where we hope to answer this concern. We have evaluated our model not only on generation tasks in standard chat benchmarks but also included probability-based tasks in LM eval same as Nvidia hybrid Mamba2. We also see that a model distilled from Llama-3 performs better than the Nvidia hybrid Mamba2 model, which was trained from scratch with 3.2T tokens. In addition we train on a larger set of tokens.

---

### Official Review · Reviewer_NYcb · 2024-07-14

**Soundness:** 1
**Presentation:** 2
**Contribution:** 2
**Rating:** 6
**Confidence:** 4

**Summary:**

The paper proposes a way to convert a Transformer architecture to a Mamba architecture by first initializing a Mamba layer based on a pre-trained attention layer in order to facilitate faster convergence followed by knowledge distillation from the Transformer to the Mamba model. The training procedure only involves post-training on task-specific data. The paper also uses DPO for the knowledge distillation and the authors emphasize the novelty of this method for knowledge distillation. Following their proposed method, the authors demonstrate how to quickly train a hybrid architecture with some attention and some Mamba layers based on existing LLMs, particularly Zephyr 7B. Finally, the paper proposes a way to perform speculative decoding and gain speed up using Mamba architectures.

**Strengths:**

The paper's writing is mostly clear and can be followed. The idea of initializing a Mamba layer based on pre-trained Transformer layers is novel and interesting. The authors evaluate the models on several benchmarks.

**Weaknesses:**

Looking at the performance in Table 1, increasing the attention percentage always improves performance in a significant way (most clear in MMLU). As such, I am not convinced that the method is actually working. There is also a lack of proper baselines. For example, no comparison is performed with random initialization. Similarly, the results for 100% Mamba architecture is not included. While this might not be as good as 50% architecture, it is still very much needed to understand the trade-off.

The idea of converting a pre-trained Transformer to Mamba is quite generic. In particular, there does not seem to be any specific scale needed to get the method to work. As such, it is not clear why the authors decided to mainly test the method on such a large scale.  For example, a small Transformer model can be trained from scratch, used to create a Mamba model and then compared in terms of language modeling benchmarks such as perplexity. Indeed this experiment is done for one experiment in Section 7 and shows poor performance of the method in terms of perplexity. As a side note, the paragraph starting in line 275 is titled Does PPL correspond to ability?. However, the discussion in that paragraph does not answer this question at all. Moreover there is actually a correspondence and we see degradation in task specific performance similar to PPL.

Moreover, the authors mention that they do not target pre-trained language model capabilities. Why is that the case?  Furthermore, the comparison is done on a hybrid architecture which further makes it confusing to determine the success of the method. Note that large models have a lot of redundancies as suggested by recent work in pruning and this makes it harder to see whether other attention layers are stepping in to make up for the poor performance of Mamba layers or something else is happening.

Overall I find the results not convincing enough and not supporting the claims made in the paper about the success of the method.

**Questions:**

1. The definition of # Gen in Figure 2 (right) is very vague and unclear. Can you please elaborate?

2. In line 182 it is mentioned that the parallel mode of Mamba is faster than a Transformer. Is there a reference for this? If so I think it would be useful to include it.

3. In Algorithm 2, the verify part may return the state from an earlier point, i.e. $h_j$ instead of $h_{k^\prime}$. How is this ok for the rest of the algorithm? This means a set of tokens will be ignored as the algorithm will never go over $j + 1, \ldots, k^\prime$ to incorporate them into the state.

**Limitations:**

The authors mention that the performance is only measured after fine-tuning on chat corpora instead of full pre-training due to resource limitations. While at 7B scale this is acceptable, as mentioned above, there is no need to focus on that scale from the start. Further limitations, such as degradation of performance when switching from attention to Mamba layers should be more deeply discussed.

---

> ### Author Rebuttal · Authors · 2024-08-07
>
> Thank you very much for your comments! We kindly ask you to find our responses below.
>
> $\mathcal{Q}.$ No comparison is performed with random initialization for task specific performance.
>
> $\mathcal{A}.$ We added a comparison with random initialization. Please refer to 3a in the general response. With the Mamba default initialization and the same training tokens, the model gets a very low score in MT-bench and AlpacaEval.
>
> $\mathcal{Q}$ Does PPL correspond to ability?. However, the discussion in that paragraph does not answer this question at all. Moreover there is actually a correspondence and we see degradation in task specific performance similar to PPL
>
> $\mathcal{A}.$ In the new ablations that we ran, we found that initializing using attention weights not only gives lower PPL, but also significantly improves task-specific performance.
>
> $\mathcal{Q}.$ Similarly, the results for 100% Mamba architecture is not included. While this might not be as good as 50% architecture, it is still very much needed to understand the trade-off.
>
> Furthermore, the comparison is done on a hybrid architecture which further makes it confusing to determine the success of the method.
>
> $\mathcal{A}.$ We include the 1/8 Mamba model in the new result in the general response.
>
> Here we also compare with 100% Mamba model (trained progressively). These results are less good than hybrid, but significantly better than random initialization.
>
> | Model                         | MT-Bench (score) | AlpacaEval (LC win %) against GPT-4       | AlpacaEval (win %) against GPT-4   | ARC              | HellaSwag                     | MMLU                           | TruthfulQA                     |
> |-------------------------------|------------------|------------------------------|--------------------------------|------------------|-------------------------------|--------------------------------|--------------------------------|
> | Zephyr-Mamba (16 attention)   | **7.31**             | **20.66$_{0.74}$**           | **16.69$_{1.10}$**             | $54.27_{1.46}$   | $75.07_{0.43}$                | $55.38_{11.86}$                | $53.60_{1.58}$                 |
> | Zephyr-Mamba (8 attention)    | 7.03             | 17.16$_{0.69}$               | 13.11$_{1.00}$                 | $50.85_{1.46}$   | $71.77_{0.45}$                | $51.19_{10.92}$                | $47.75_{1.55}$                 |
> | Zephyr-Mamba (4 attention)    | 6.40             | 15.32$_{0.66}$               | 12.96$_{1.02}$                 | $50.09_{1.46}$   | $68.11_{0.45}$                | $48.44_{10.06}$                | $46.16_{1.56}$                 |
> | Zephyr-Mamba (0 attention)    | 5.20             | 7.39$_{0.39}$                | 6.92$_{0.77}$                  | $48.38_{1.46}$   | $66.54_{0.47}$                | $35.50_{6.23}$                 | $44.58_{1.52}$                 |
>
> $\mathcal{Q}.$ The authors mention that they do not target pre-trained language model capabilities. Why is that the case?
>
> $\mathcal{A}.$ We apologize for the confusion in our writing. We do think this approach captures general *pretraining* ability. To clarify, given a limited amount of distillation compute, we hypothesized that the most effective usage of compute is to distill on chat data, which is high-quality and task relevant.
>
> Even though we train on chat data, we want to emphasize that our model has strong performance on general benchmarks. For example, our model distilled from Llama-3 performs better than the NVIDIA hybrid Mamba2 model on general benchmarks. Please refer to 2 in the general response.
>
> $\mathcal{Q}.$ Since there is redundancy in Transformer layers, are the Mamba layers actually useful?
>
> $\mathcal{A}.$ This is a good question, and we were curious as well. Please refer to 3b in our general response. This result confirms that adding Mamba layers is very critical and that the results are not just from the remaining attention.
>
> $\mathcal{Q}.$ The definition of # Gen in Figure 2 (right) is very vague and unclear. Can you please elaborate?
>
> $\mathcal{A}.$ This is the number of tokens we generate at each generation step. It is calculated as the average number of draft tokens we accept at each iteration, plus an additional token which we is sampled from the target model, as per the speculative decoding algorithm [1].
>
> $\mathcal{Q}.$ In line 182 it is mentioned that the parallel mode of Mamba is faster than a Transformer. Is there a reference for this? If so I think it would be useful to include it.
>
> $\mathcal{A}.$ It is more efficient for long sequences because it doesn't require $O(L^2)$ computation. Please refer to the left figure in Figure 8 of the [Mamba](https://arxiv.org/pdf/2312.00752) paper.
>
> $\mathcal{Q}.$ In Algorithm 2, the verify part may return the state from an earlier point, i.e. $h_j$ instead of $h_{k'}$. How is this ok for the rest of the algorithm? This means a set of tokens will be ignored as the algorithm will never go over $j+1, ..., k$ to incorporate them into the state.
>
> $\mathcal{A}.$ We are sorry for the confusion.
> Yes you are correct in that in speculative coding we may throw away generated tokens.
> In the case where a state from an earlier point is returned by the verify part, the draft model will also resume decoding from a previous snapshot of the state, recomputing the states for the accepted tokens from its cache.
>
> ---
>
> [1] Fast Inference from Transformers via Speculative Decoding (https://arxiv.org/abs/2211.17192)

---

> > ### Comment · Reviewer_NYcb · 2024-08-11
> >
> > Dear Authors,
> >
> > Thank you very much for the additional experiments and your clarifications and replies. I have asked for some additional clarifications below.
> >
> > >  In the new ablations that we ran, we found that initializing using attention weights not only gives lower PPL, but also significantly improves task-specific performance.
> >
> > I could not find the new PPL results. Could you please share them as well?
> >
> > > About Random Init: We added a comparison with random initialization. Please refer to 3a in the general response. With the Mamba default initialization and the same training tokens, the model gets a very low score in MT-bench and AlpacaEval.
> > > Regarding no Mamba layers:  This is a good question, and we were curious as well. Please refer to 3b in our general response. This result confirms that adding Mamba layers is very critical and that the results are not just from the remaining attention.
> >
> > Thank you for providing these results. Just to confirm, you still keep the FFN parts of the layers, and only remove the attention, then do fine-tuning? I think if this is the case, this is an important result as it shows the efficacy of using Mamba blocks and the projection. Combined with the obtained speedup I think this makes the results quite interesting. However, the trend that using Mamba blocks lead to decreased accuracy is completely observable in all the results. As such, I think it is paramount that this limitation be clearly specified and discussed in the paper. Do authors agree on this or have I misinterpreted the results?
> >
> > >  We are sorry for the confusion. Yes you are correct in that in speculative coding we may throw away generated tokens. In the case where a state from an earlier point is returned by the verify part, the draft model will also resume decoding from a previous snapshot of the state, recomputing the states for the accepted tokens from its cache.
> > Could you please fix this in the next revision of the paper? As I think the current Algorithm is incorrectly specified and does not include the recomputation part which can be confusing.
> >
> > Assuming the above are addressed, I will be increasing my score as I think the new results provide enough supporting evidence that the method is working.

---

> ### Author Response · Authors · 2024-08-11
>
> We thank again the reviewer for the helpful feedback.
>
> $\mathcal{Q}.$ I could not find the new PPL results. Could you please share them as well?
>
> $\mathcal{A}.$ Since our models are alignment using DPO, we evaluated ppl of our final models on lambada_openai. Here are ppl of models in the ablation results.
>
> | Model                         | PPL |
> |-------------------------------|-----|
> | Hybrid Mamba (50%) w/o Attention init |  55.01 |
> | Hybrid Mamba (50%) w Attention init  | 6.20  |
>
> | Model                         | PPL |
> |-------------------------------|-----|
> | 50% Attention w/o Mamba Block | 151.98 |
> | 50% Attention w Mamba Block   | 6.20  |
>
> $\mathcal{Q}.$ Just to confirm, you still keep the FFN parts of the layers, and only remove the attention, then do fine-tuning? I think if this is the case, this is an important result as it shows the efficacy of using Mamba blocks and the projection. Combined with the obtained speedup I think this makes the results quite interesting. However, the trend that using Mamba blocks lead to decreased accuracy is completely observable in all the results. As such, I think it is paramount that this limitation be clearly specified and discussed in the paper. Do authors agree on this or have I misinterpreted the results?
>
> $\mathcal{A}.$ Yes, we still keep those three Gated MLPs (FFN components) and only remove half of the attention layers before fine-tuning. We acknowledge that changing attention to Mamba may still lead to decreased performance as more Mamba layers are introduced. At the same time, we believe that as computational resources scale up, this impact will be mitigated. Additionally, the inductive bias learned from attention can help Mamba scale up, as our model has already outperformed other hybrid Mamba models trained from scratch with significantly more tokens. We definitely will address and discuss this limitations in the paper.
>
> Additionally, we will update the speculative decoding section to improve and correct the explanation of the algorithm.

---

> > ### Comment · Reviewer_NYcb · 2024-08-13
> >
> > Dear Authors,
> >
> > Thank you for the additional results.
> >
> > One of my concerns remain and I think it would be a great addition if the authors could show whether the model relies on the language models being large scale or is applicable on smaller scales as well. As I mentioned originally in my review, it is possible to train a smaller transformer from scratch for a reasonable number steps, then do the same approach for converting layers to Mamba layers and compare the two models in terms of perplexity (or any other metric). Currently the paper is only evaluating on 7B models but it is not clear whether there is a limitation for smaller sizes or not. Either way I think it should be discussed and would be a nice addition.
> >
> > Still, given the new results provided in the rebuttal, I think there is sufficient evidence to show the method has merit and can be useful. As such, I have increased my score.

---

> > > ### Author Response · Authors · 2024-08-14
> > >
> > > We deeply appreciate all the supportive and constructive feedback. We are running a smaller scale (110M) ablation experiment to address your concerns. We will definitely discuss this and include it in the next version.

---

### Author Rebuttal · Authors · 2024-08-07

We appreciate each reviewer's insightful feedback. The excellent feedback has helped us improve our work and obtain new experimental results. We hope these clarify the main questions from the reviews.

1. **Scaling up the traning data to 20B tokens, adding Llama-3, 1/8 attention**: We include new results as requested. We see significant improvements from more data:

Comparison between different hybrid Mamba models distilled from the teacher model Zephyr on Chat Benchmark and OpenLLMLeaderboard.

| Model | MT-Bench (score) | AlpacaEval (LC win \%) against GPT-4 | AlpacaEval (win \%) against GPT-4 |
|-|-|-|-|
| Zephyr  | **7.34** | $13.20_{0.96}$  | $10.99_{0.96}$ |
| Zephyr-Mamba (½ attention) - Old (Token 2B)     | 6.69  | $14.11_{1.01}$      | $12.60_{1.01}$     |
| Zephyr-Mamba (½ attention)     | **7.34**   | **$20.66_{0.74}$**     | **$16.69_{1.10}$**       |
| Zephyr-Mamba (¼ attention)     | 7.03        | $17.16_{0.69}$          | $13.11_{1.00}$           |
| Zephyr-Mamba (⅛ attention)     | 6.40         | $15.32_{0.66}$          | $12.96_{1.02}$           |

| Model | ARC-C  | HellaSwag | MMLU | TruthfulQA  |
|-|-|-|-|-|
| Zephyr                         | 62.03            | 84.52                         | 61.44                          | 57.44                          |
| Zephyr-Mamba (½ attention) - Old (Token 2B)     | $49.15_{1.46}$   | $75.07_{0.43}$                | $47.98_{10.21}$                | $46.67_{1.51}$                 |
| Zephyr-Mamba (½ attention)     | $54.27_{1.46}$   | $75.07_{0.43}$                | $55.38_{11.86}$                | $53.60_{1.58}$                 |
| Zephyr-Mamba (¼ attention)     | $50.85_{1.46}$   | $71.77_{0.45}$                | $51.19_{10.92}$                | $47.75_{1.55}$                 |
| Zephyr-Mamba (⅛ attention)     | $50.09_{1.46}$   | $68.11_{0.45}$     | $48.44_{10.06}$                | $46.16_{1.56}$                 |


Comparison between different hybrid Mamba models distilled from the teacher model Llama3 on Chat Benchmark.

| Model                          | MT-Bench (score) | AlpacaEval (LC win \%) against GPT-4  | AlpacaEval (win \%) against GPT-4   |
|-|-|-|-|
| Llama 3-8b-instruct         | 7.90       | 22.90   | 22.6   |
| Llama 3-Mamba (½ attention) | 7.50       | 22.92   | 20.42  |
| Llama 3-Mamba (¼ attention) | 6.80       | 16.69   | 15.64 |
| Llama 3-Mamba (⅛ attention) | 6.51       | 15.02   | 13.03 |
| Llama 3-Mamba2 (½ attention) | 7.43      | 23.01   | 21.76 |

2. **Using Mamba-2 and comparison to [Nvidia Hybrid Mamba-2](https://arxiv.org/abs/2406.07887)**:
As requested we extend our approach to Mamba-2, a recently released variant. We also compare to Nvidia Hybrid Mamba-2 a from scratch pretrained Mamba-2 and Attention model. We compare our model with this model on standard benchmarks and see comparable results.

| Model                     | Training Tokens | WG    | PIQA  | HellaSwag | ARC-Easy  | ARC-Challenge  | MMLU  | OpenBook | TruthFul | PubMed | RACE  | Avg   |
|-|-|-|-|-|-|-|-|-|-|-|-|-|
| Nvidia Hybrid Mamba-2  | PT-3.5T            | **71.27** | 79.65 | 77.68  | 77.23  | 47.70         | 53.60 | 42.80    | 38.72   | 69.80  | 39.71 | 59.82 |
| Llama3-Mamba  (½ attention)           | FT-20B             | 68.43  | 78.02  | 77.55     | 72.60      | 50.85     | **59.26** | **44.00**  | 45.24  | 72.80  | 38.47 | 60.72 |
| Llama3-Mamba  (¼ attention) | FT-20B            | 68.21  | 78.23 | 76.59    | 72.56     | 49.60         | 55.20 | 43.10    | 45.01   | 70.10  | 38.31 | 59.69 |
| Llama3-Mamba2  (½ attention)      | FT-20B          | 69.38    | **80.74** | **78.08**     | **77.40**     | **56.06**         | 56.09  | 43.40     | **53.47**    | **73.20**   | **41.24** | **62.91** |

3. **Requested Ablation Studies**:

Reviewers requested additional ablation studies. We include them here.

3a. **Mamba Initialization**:

Default random initialization versus Linear Projection from Attention. (Trained using the same recipe).

| Model                                             | MMLU  | ARC-C | TruthfulQA | HellaSwag | MT-Bench | AlpacaEval LC win against GPT-4 |
|-|-|-|-|-|-|-|
| Hybrid Mamba (50%) w/o Attention init | 26.21  | 25.26 | 34.01      | 27.91     | 1.04        | 0.02                               |
| Hybrid Mamba (50%)    | 47.98  | 49.15 | 46.67     | 75.07     | 6.69     | 14.11                           |

These results confirm that the initialization from attention weights is critical.

3b. **Necessity of Mamba layers**:

Mamba blocks versus only keeping subset of attention. (Same training procedure)

| Model               | MMLU  | ARC-C | TruthfulQA | HellaSwag | MT-Bench | AlpacaEval LC win against GPT-4 |
|-|-|-|-|-|-|-|
| 50% Attention w/o Mamba Block     | 24.46 | 21.93 | 32.39      | 27.91     | 1.01     | 0                               |
| 50% Attention w Mamba Block       | 47.98 | 49.15 | 46.67      | 75.07     | 6.69     | 14.11                           |

These results confirm that adding Mamba layers is critical and the performance is not just compensated by the remaining attention layers.

4. **Optimized speculative decoding implementation for Mamba**. This is achieved by fusing the kernels that perform the recomputation of previous states from the cache, with the kernels that perform the decoding of the next sequence of draft tokens. This is done both for the convolutional part and for the SSM part of the Mamba block.  Additionally, we now include results for Mamba 7B, for which we train a Llama-3, 4-layer, 1B parameters speculator.

### Table 1: Speedup on Mamba 7B

| **k Value** | **# Gen tokens** | **Speedup** |
|-|-|-|
| k=3         |3.19         |2.1x              |
| k=4         |3.56         |2.1x              |

### Table 2: Speedup on Mamba 2.8B (Optimized vs. Old Results)
| **k Value** | **Speedup (Optimized)** | **Speedup (Old)** |
|-|-|-|
| k=3         | **2.3x**                    | 1.49x             |
| k=4         | **2.6x**                   | 1.56x

---

### Decision · Program_Chairs · 2024-09-25

**Decision:**

Accept (poster)

**Comment:**

Reviewers all agreed that the initial version of the paper had inadequate experimentation, and some found the exposition inadequate.  It is clear from the revised version however that these deficiencies have been largely corrected, and although the reviewer scores remain numerically low, the revised paper will be a valuable contribution to the conference.